# Algorithmic bias in social research: A meta-analysis

**Alrik Thiem** [1]*, **Lusine Mkrtchyan**[1], **Tim Haesebrouck** [2], **David Sanchez**[3]

**1** Faculty of Humanities and Social Sciences, University of Lucerne, Lucerne, Switzerland, **2** Institute for International Studies, Ghent University, Ghent, Belgium, **3** LINK Institute, Lucerne, Switzerland

* alrik.thiem@unilu.ch

## Abstract

Both the natural and the social sciences are currently facing a deep "reproducibility crisis". Two important factors in this crisis have been the selective reporting of results and methodological problems. In this article, we examine a fusion of these two factors. More specifically, we demonstrate that the uncritical import of Boolean optimization algorithms from electrical engineering into some areas of the social sciences in the late 1980s has induced *algorithmic bias* on a considerable scale over the last quarter century. Potentially affected are all studies that have used a method nowadays known as Qualitative Comparative Analysis (QCA). Drawing on replication material for 215 peer-reviewed QCA articles from across 109 high-profile management, political science and sociology journals, we estimate the extent this problem has assumed in empirical work. Our results suggest that one in three studies is affected, one in ten severely so. More generally, our article cautions scientists against letting methods and algorithms travel *too* easily across disparate disciplines without sufficient prior evaluation of their suitability for the context in hand.

## Introduction

Both the natural and the social sciences are currently facing a deep "reproducibility crisis" [1–4]. When asked what factors contribute to irreproducible research, more than 40 percent of 1573 scientists surveyed by *Nature* in 2016 responded that problems with "methods" often or always contribute, and almost 70 percent that "selective reporting" often or always contributes [5]. What, however, when it is not scientists themselves who, consciously or unconsciously, distort objective reporting, but the methods they employ? This may happen, for example, when the algorithms implemented in data analysis software are predisposed towards limiting their search only to specific regions of the full solution space or when an algorithm is transferred to a context other than the one which it was originally developed in without adequate prior assessment of the consequences of such transfers [6–8]. These problems have so far been given relatively short shrift in the current debate, most likely because of their rather technical nature. However, they may be one of the most significant contributors to irreproducibility.

In this article, we reveal a specific case of algorithmic bias in research that has relied on the method of Qualitative Comparative Analysis (QCA). More specifically, we demonstrate that

**Data Availability Statement:** Full replication material is available at osf.io/qwn7g.

**Funding:** The Swiss National Science Foundation has generously funded this research under grant award number PP00P1_170442 to AT. URL:

http://www.snf.ch The funder had no role in study design, data collection and analysis, decision to publish, or preparation of the manuscript.

**Competing interests:** The authors have declared that no competing interests exist.

the uncritical import of Boolean optimization algorithms from electrical engineering into causal data analysis with QCA has generated such bias through a transfer of context. A specific objective function that ensures an algorithm's identification of a minimal cost solution when designing electrical switching circuits is responsible for this effect. While this function is very useful in electrical engineering applications, it is not suited for use in QCA because rival models of a certain architecture are systematically suppressed. Among the suppressed models, however, may be the one representing the true data-generating process. Drawing on replication material for 215 peer-reviewed articles from across 109 high-profile management, political science and sociology journals that have employed QCA, we measure the extent this problem has assumed in empirical work. Our results suggest that one in three studies is affected, one in ten severely so. More generally, our article cautions scientists against letting methods and algorithms travel *too* easily across disparate disciplines without sufficient prior evaluation of their suitability for the context in hand.

## QCA: Foundations, diffusion, analytical approach

QCA is a so-called configurational method for causal inference introduced in the 1980s by US sociologists Kriss Drass and Charles Ragin, who sought to bridge the gulf between variable-oriented and case-oriented research at a time when the "paradigm wars" were reaching another peak in many fields of the social sciences [9, 10]. Yet, as any other method of observational data analysis, QCA did not simply fall out of the sky. Unlike methods developed on the basis of counterfactual, interventionist, mechanistic or probabilistic theories of causation, QCA firmly rests on the regularity theory of INUS causation [11–14]. Although regularity theories were long relegated to the margins of research on causation and causal inference, numerous areas of the natural as well as the social sciences, from economics over neurology to psychology, continued to invoke the notion of INUS conditionality for building causal arguments [15–21].

By importing the Quine-McCluskey algorithm (QMC) from electrical engineering into causal inference with QCA, the major accomplishment of Drass and Ragin was to solve the so-called problem of epiphenomenalism—one of the most vexing problems that had until then plagued the INUS theory [22]. With the removal of this stumbling block, interest in regularity theories and INUS causation started to grow again among both philosophers of science and methodologists [11, 23].

Introductions to QCA generally portray the method's main analytical principle as follows: if two empirical cases exhibit identical values on an endogenous factor as well as all exogenous factors apart from a single one, then it can be deduced that the one factor on which these two cases differ cannot be ascribed causal relevance to in the context of the remaining factors because the outcome shows no change. Although different algorithms have been implemented in software over the years for conducting research with QCA, of which QMC was only the first, the vast majority of social scientists still take QMC to be QCA's formal heart [24–27]. Yet surprisingly, technically adequate expositions of QMC hardly exist in the social-scientific literature on QCA. Due to its central place and its relevance for our ensuing argument, it is therefore essential that we introduce this algorithm's procedural protocol in as much detail as necessary. At the same time, we emphasize that our discussion is not limited to QMC, but extends to *all* optimization algorithms currently in use in configurational data analysis or of potential use for such purposes (cf. [28, 29]). In this connection, our argument also updates [30] and [31] insofar as we hold that it is the objective function given to an optimization algorithm, not the algorithm *per se*, that may induce bias.

## Switching circuit optimization with QMC: A primer

Switching circuits provide basic building blocks for the design of many digital systems that have made our modern world possible. The mathematical framework for analyzing the conversion of a given set of input signals to a desired set of output signals in order to make a circuit perform a specific function is provided by the algebra of switching circuits and logic design, a specific variant of Boolean algebra [32]. In this connection, one of the most important questions electrical engineers have to solve concerns the minimization of a circuit's hardware costs. Given two different circuits that produce the same output when provided with the same input, the circuit that is cheaper to build is preferred. More formally, this question can be phrased as follows: Given a switching function $f$ and an objective function $\mathcal{F}$ defined on the set of $f$-equivalent switching functions $S_f$, what is the set of $f$-equivalent switching functions $S_f^*$ for which $\mathcal{F}$ reaches a minimum?

There are many ways in which $\mathcal{F}$ can be defined. It could relate to the number of circuit gates, the number of gate contacts, or a more complex requirement of the form $aP + bQ + cR$, where $P$, $Q$, and $R$ represent the number of gates of a certain type and $a$, $b$ and $c$ are weighting coefficients on unit price, reliability or other economical or technical criteria [33]. For example, consider the circuit presented in panel (a) of Fig 1, whose switching function is given by $(W' + X)\cdot(W' + Y) = Z$, and whose corresponding function table is presented in the left and middle parts of panel (b) of Fig 1. As is customary in writing logic functions for switching circuits, "'" denotes the logical *NOT-operator* (open contact), "+" the logical *OR-operator* (switches in parallel), "·" the logical *AND-operator* (switches in series), and "=" the logical ONLY-IF-THEN-operator (no separate symbol). For convenience, "·" will be dropped if there is no risk of confusion, and AND will take precedence over OR. A function whose main operation is an OR-operation is called a *sum of products* (SOP), a function whose main operation is an AND-operation is called a *product of sums* (POS).

The output variable, $Z$, takes on the value 1 for combinations $W'X'Y'$, $W'X'Y$, $W'XY'$, $W'XY$ and $WXY$; and it takes on the value 0 for combinations $WX'Y'$, $WX'Y$ and $WXY'$ of the input variables $W$, $X$ and $Y$. Combinations for which some switching function $f$ equals 1 are called positive minterms, those for which $f$ takes on the value 0 are called negative minterms.

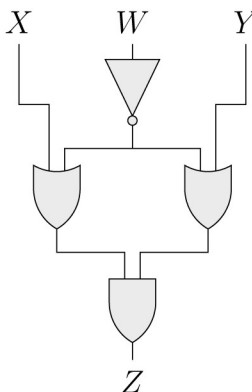

| (a) | | | | (c) |
|---|---|---|---|---|
| $Z$ | $W$ | $X$ | $Y$ | $Z$ |
| 1 | 0 | 0 | 0 | 1 |
| 1 | 0 | 0 | 1 | 1 |
| 1 | 0 | 1 | 0 | 1 |
| 1 | 0 | 1 | 1 | 1 |
| 0 | 1 | 0 | 0 | 0 |
| 0 | 1 | 0 | 1 | 0 |
| 0 | 1 | 1 | 0 | 0 |
| 1 | 1 | 1 | 1 | 1 |

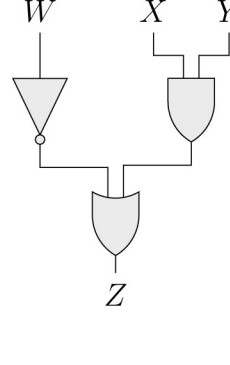

**(a)** $(W' + X)\cdot(W' + Y) = Z$  **(b)** Function Table for (a) and (c)   **(c)** $W' + X \cdot Y = Z$

**Fig 1. Two equivalent switching functions and their switching circuits.** (a): $(W' + X) \cdot (W' + Y) = Z$. (b): Function Table for (a) and (c). (c): $W' + X \cdot Y = Z$.

Furthermore, every appearance of a variable in *f* is called a literal. To realize the given switching function as a circuit, four gates are required: one NOT-gate, two OR-gates and one AND-gate.

Now consider the circuit presented in panel (c) of Fig 1, whose switching function is given by $W' + XY = Z$, and whose corresponding function table is presented in the middle and right parts of panel (b) of Fig 1. As *Z* takes on the same values for the same minterms as $(W' + X)$ $(W' + Y) = Z$ does, the two functions are equivalent: they behave identically for the same combinations of input conditions. However, instead of four gates, $W' + XY = Z$ requires only three gates: one NOT-gate, one AND-gate and one OR-gate. Additionally, it is has only three literals instead of four. The hardware costs of the circuit in panel (c) are thus lower than those incurred by the circuit in panel (a). Is there a systematic way of identifying the cheaper circuit in panel (c)? One possibility to get from $(W' + X)(W' + Y) = Z$ to $W' + XY = Z$ is by applying the Boolean-algebraic law of distribution with respect to OR, according to which $\Phi + \Psi\Omega = (\Phi + \Psi)(\Phi + \Omega)$. Contrast this with linear algebra, where "+" (plus) does not distribute over "·" (times) [34]. By reversing the distribution, we can architect a circuit that behaves exactly as the one represented in panel (a) of Fig 1 yet at lower hardware costs.

However, the optimization of a switching circuit by means of applying Boolean-algebraic laws to its corresponding switching function becomes increasingly cumbersome as the complexity of this circuit's input-output specification increases. To circumvent this problem, an algorithmic alternative was proposed by Edward McCluskey, who improved on earlier work carried out by Claude Shannon and Willard Quine at the intersection of electrical engineering and analytical philosophy, in the form of QMC [35–40]. Because of its accessibility to beginning students of electrical engineering, all elementary textbooks on applications of Boolean algebra, switching circuit theory or logic design contain sections on this algorithm [32, 41–43]. A simple example shall be provided in the following. Without any loss of generality, we will adopt as our objective function the identification of a SOP function that has the minimum number of AND-gates among all equivalent functions. We call such a function a minimal sum.

Consider a new switching function, presented in Table 1, which involves four input variables, *V*, *W*, *X* and *Y*, and the output variable *Z*. In this table, each minterm now also has an ID character, which is the decimal number associated with the binary number representation of the respective minterm. For instance, the ID of the second row of Table 1 is 1 because the number 1 is expressed as $0 \times 2^3 + 0 \times 2^2 + 0 \times 2^1 + 1 \times 2^0$ under a binary number system; the ID of the row before the last is 14 because $1 \times 2^3 + 1 \times 2^2 + 1 \times 2^1 + 0 \times 2^0 = 14$. The last row groups together all minterms whose respective combination of values under *V*, *W*, *X* and *Y* is

**Table 1. Function table.**

| ID | V | W | X | Y | Z |
|---|---|---|---|---|---|
| 0 | 0 | 0 | 0 | 0 | 1 |
| 1 | 0 | 0 | 0 | 1 | 1 |
| 3 | 0 | 0 | 1 | 1 | 1 |
| 4 | 0 | 1 | 0 | 0 | 0 |
| 6 | 0 | 1 | 1 | 0 | 0 |
| 8 | 1 | 0 | 0 | 0 | 1 |
| 9 | 1 | 0 | 0 | 1 | 0 |
| 11 | 1 | 0 | 1 | 1 | 0 |
| 13 | 1 | 1 | 0 | 1 | 0 |
| 14 | 1 | 1 | 1 | 0 | 1 |
| 2,5,7,10,12,15 | . . . | . . . | . . . | . . . | – |

**Table 2. Derivation of prime implicants.**

| (a) | | | | | | (b) | | | | | | (c) | | | | | |
|---|---|---|---|---|---|---|---|---|---|---|---|---|---|---|---|---|---|
| | V | W | X | Y | | | V | W | X | Y | | | V | W | X | Y | |
| 0 | 0 | 0 | 0 | 0 | ✓ | 0,1 | 0 | 0 | 0 | – | ✓ | 0,1,2*,3 | 0 | 0 | – | – | ✗ |
| 1 | 0 | 0 | 0 | 1 | ✓ | 0,2* | 0 | 0 | – | 0 | ✓ | 0,2*,8,10* | – | 0 | – | 0 | ✗ |
| 2* | 0 | 0 | 1 | 0 | ✓ | 0,8 | – | 0 | 0 | 0 | ✓ | 1,3,5*,7* | 0 | – | – | 1 | ✗ |
| 8 | 1 | 0 | 0 | 0 | ✓ | 1,3 | 0 | 0 | – | 1 | ✓ | 8,10*,12*,14 | 1 | – | – | 0 | ✗ |
| 3 | 0 | 0 | 1 | 1 | ✓ | 1,5* | 0 | – | 0 | 1 | ✓ | | | | | | |
| 5* | 0 | 1 | 0 | 1 | ✓ | 2*,3 | 0 | 0 | 1 | – | ✓ | | | | | | |
| 10* | 1 | 0 | 1 | 0 | ✓ | 2*,10* | – | 0 | 1 | 0 | ✓* | | | | | | |
| 12* | 1 | 1 | 0 | 0 | ✓ | 8,10* | 1 | 0 | – | 0 | ✓ | | | | | | |
| 7* | 0 | 1 | 1 | 1 | ✓ | 8,12* | 1 | – | 0 | 0 | ✓ | | | | | | |
| 14 | 1 | 1 | 1 | 0 | ✓ | 3,7* | 0 | – | 1 | 1 | ✓ | | | | | | |
| 15* | 1 | 1 | 1 | 1 | ✓ | 5*,7* | 0 | 1 | – | 1 | ✓* | | | | | | |
| | | | | | | 10*,14 | 1 | – | 1 | 0 | ✓ | | | | | | |
| | | | | | | 12*,14 | 1 | 1 | – | 0 | ✓ | | | | | | |
| | | | | | | 7*,15* | – | 1 | 1 | 1 | ✓* | | | | | | |
| | | | | | | 14,15* | 1 | 1 | 1 | – | ✗ | | | | | | |

not relevant for the specification of the circuit because these combinations do not occur; hence the dash, "—", under $Z$. In switching circuit theory, such functions are called incompletely specified functions, and the non-specified combinations are referred to as don't care minterms or simply don't cares. All information contained in the function table in Table 1 can be more compactly summarized as $Z(V, W, X, Y) = \sum m_1 (0, 1, 3, 8, 14) + \sum m_d (2, 5, 7, 10, 12, 15)$, where $m_1$ stands for the positive minterms and $m_d$ for the don't cares. This expression represents the canonical sum.

The first algorithmic phase of QMC consists in eliminating as many literals as possible from the positive minterms by making use of a successive combination of three Boolean-algebraic laws: the dual to the distributive law used above in relation to the circuits shown in Fig 1, according to which $\Phi(\Psi + \Omega) = \Phi\Psi + \Phi\Omega$; the law of complementarity, according to which $\Phi + \Phi' = 1$, and the law of identity, which states that $\Phi \cdot 1 = \Phi$. To this end, QMC arranges all positive minterms as well as all don't cares in blocks of minterms for which the number of 1s they contain is the same. This is shown in sub-table (a) of Table 2. The asterisk marks off don't cares from positive minterms. For example, the second block includes minterms 1, 2* and 8 because they are the only ones that contain a single 1 in their binary number representation. As there is only one possibility for having not a single 1, as many 1s as there are input variables, respectively, minterms 0 and 15* each have their own block.

Provided the difference in the number of 1s in adjacent blocks is one, minterms from adjacent blocks whose IDs differ by a number that can be expressed as a power of 2 can be reduced by eliminating literals on the basis of the three Boolean-algebraic laws introduced above. These reduced minterms are called (proper) implicants. For example, minterms 0 and 1 can be combined because $1 - 0 = 1$, which can be expressed as $2^0$. Specifically, literals $Y'$ and $Y$ can be eliminated because $V'W'X'Y' + V'W'X'Y$ equals $V'W'X'(Y' + Y)$ by distribution, which in turn equals $V'W'X'(1)$ by complementarity, which in turn equals $V'W'X'$ by identity.

To indicate that a minterm has been used in creating an implicant, it receives a tick mark, and the newly formed implicant is transferred to a new table, sub-table (b) in Table 2, where the ID of a new implicant is simply the combination of IDs from the minterms that have gone into creating it. The eliminated literal is marked by a dash. Each minterm can be used more

than once in deriving implicants because of the law of idempotency, according to which $\Phi + \Phi = \Phi$. This process continues until no further eliminations are possible. From a table with 11 minterms, QMC has progressed to a table with 15 implicants. Seemingly, the situation has become more complex rather than less complex, but further reductions are possible.

In sub-table (b), implicants from adjacent blocks can be combined if the differences in their respective double IDs is not only the same, but also again a number that can be expressed as a power of two. For example, implicant (0, 1) cannot be combined with implicant (1, 3) because the powers of two that make up the difference in their IDs are not the same; $1 - 0 = 1$, but $3 - 1 = 2$. In contrast, implicant (0, 1) can be combined with implicant ($2^*$, 3) because the differences in their double IDs are both 2, which is a number that can be written as a power of two, namely $2^1$. Specifically, literals $X'$ and $X$ can be eliminated from (0, 1) and ($2^*$, 3) because $V'W'X' + V'W'X = V'W'(X' + X) = V'W'(1) = V'W'$. As before, to indicate that an implicant has been used in creating another, shorter implicant it receives a tick mark, and the newly formed implicant is transferred to a new table, sub-table (c) in Table 2. Also, the ID character of a newly formed implicant is simply the combination of IDs from the implicants that have gone into creating it. This process continues until no further reductions are possible.

In sub-table (b), implicant (14, $15^*$) could not be further minimized to yield a shorter implicant, and none of the implicants in sub-table (c) can be reduced further. Implicants that cannot be reduced further are called prime implicants and receive a cross instead of a tick mark. Once there are only crosses left, QMC stops the process of eliminating literals. Theoretically, all prime implicants could now simply be joined to yield the complete sum, $Z = VWX + V'W' + W'Y' + V'Y + VY'$, which is a correct representation of the canonical sum. As this function corresponds to a switching circuit with as many AND-gates as that associated with the canonical sum yet fewer literals, less hardware would be required. Yet, since our objective function requires a SOP function with as few AND-gates as possible, we need to test whether some prime implicants can be eliminated without losing equivalence. What is thus needed is not a complete sum but an irredundant sum, that is, a SOP function of prime implicants that does not contain any unnecessary prime implicants. In a second phase of optimization, QMC thus identifies those functions that have the minimum number of prime implicants, yet still behave exactly as specified in Table 1.

By invoking the consensus theorem, according to which $\Phi\Psi + \Phi'\Omega + \Psi\Omega = \Phi\Psi + \Phi'\Omega$, we immediately see that, for example, $W'Y'$ is redundant in the presence of $V'W'$ and $VY'$. As before, however, the direct application of theorems of Boolean algebra can become very cumbersome. QMC therefore decomposes the complete sum via a device called prime implicant chart, in which all prime implicants are listed along the rows and all positive minterms along the columns. The prime implicant chart that results from the first step of minimization performed in Table 2 is shown in Table 3. Don't cares that have been used in deriving an implicant are not listed alongside the positive minterms, nor would be prime implicants that had

**Table 3. Prime implicant chart for function table in Table 1.**

| | | Minterms | | | | |
|---|---|---|---|---|---|---|
| | | 0 | 1 | 3 | 8 | 14 |
| | | $V'W'X'Y'$ | $V'W'X'Y$ | $V'W'XY$ | $VW'X'Y'$ | $VWXY'$ |
| $1,3,5^*,7^*$ | $V'Y$ | – | × | × | – | – |
| $8,10^*,12^*,14$ | $VY'$ | – | – | – | × | × |
| $0,1,2^*,3$ | $V'W'$ | × | × | × | – | – |
| $0,2^*,8,10^*$ | $W'Y'$ | × | – | – | × | – |
| $14,15^*$ | $VWX$ | – | – | – | – | × |

been derived solely from don't cares. If a prime implicant covers a positive minterm, it receives a cross in the chart, if it does not, it receives a dash. For instance, $VY'$ covers $VWXY'$ because all the former's literals form a subset of all the latter's. Any combination of prime implicants that covers all positive minterms yields a function equivalent to the original function, but not necessarily a function that satisfies $\mathcal{F}$.

One way of finding the switching function(s) with the fewest prime implicants would be to systematically identify all equivalent switching functions at once, and to then select the function that has the smallest number of prime implicants. Clearly, such an approach would be highly inefficient because all the objective function requires is a single switching function for which it is guaranteed that it contains the lowest possible number of prime implicants. So as to optimize the second step of minimization, QMC therefore implements a routine based on the concept of row dominance: A row $i$ in a prime implicant chart dominates another row $j$ of that chart if, and only if, $i$ has a cross in all columns in which $j$ has a cross and $i$ has a cross in at least one column in which $j$ does not have a cross [36]. As dominated prime implicants can never outperform dominating ones when the objective function specifies the minimum number of AND-gates, a solution consisting only of dominating prime implicants is guaranteed to satisfy $\mathcal{F}$.

In the prime implicant chart in Table 3, $VY'$ dominates $VWX$, $V'W'$ dominates $V'Y$, and no other prime implicant dominates any other. As $VY'$ and $V'W'$ together already cover all positive minterms, the solution that minimizes circuit hardware costs when the objective function defines the minimum number of AND-gates is given by $V'W' + VY' = Z$. Without ever identifying other irredundant sums, we already know that no other combination of prime implicants will provide an alternative to $V'W' + VY'$. Hence, the application of row dominance permits an efficient identification of irredundant sums that will also be minimal sums.

## Algorithmic bias

QCA studies usually report only a single explanatory model for their data. In other words, QCA research almost always seems to produce strong results. However, we hold that such results have often neither been the upshot of the collection of high-quality data material nor the skilful use of pertinent theories but simply an algorithmic corollary of the uncritical import of QMC from electrical engineering. To do this, let us leave the mechanics of QMC introduced in the previous section aside for the moment, and jump to the daily business of social scientist John Doe, who has collected data on some phenomenon $S$, which Doe presumes to be an effect of some combination of variables $C$, $R$, $L$ and $E$. These data, which exist for 16 cases, $c_1$ to $c_{16}$, are presented in Table 4.

What Doe does not know: social nature has determined the data-generating structure (DGS) $C'E + R'E' + CRL = S$, which has given rise to Doe's data, and which Doe seeks to uncover, or at least get closer to, as much as his data permit. Put verbally, $S$ takes on the value 1 if, and only if, $C$ takes on the value 0 and $E$ takes on the value 1 (cases $c_{10}$ and $c_{11}$), or $R$ takes on the value 0 and $E$ takes on the value 0 (cases $c_4$, $c_5$, $c_6$ and $c_{12}$), or $C$ takes on the values 1 and $R$ takes on the value 1 and $L$ takes on the value 1 (case $c_7$). That each of these conjunctions is minimally sufficient for $S$ can easily be verified: neither $C'$ nor $E$ alone are sufficient for $S$ because cases $c_{10}$ to $c_{12}$, and cases $c_1$ to $c_3$, $c_8$ and $c_9$, respectively, are associated with $S = 0$; neither $R'$ nor $E'$ alone are sufficient for $S$ because cases $c_1$ to $c_3$, and cases $c_{13}$ to $c_{16}$, respectively, are associated with $S = 0$; and neither the conjunction of $CR$, nor that of $CL$, nor that of $RL$ alone are sufficient for $S$ because cases $c_8$ and $c_9$, case $c_1$, and cases $c_{13}$ and $c_{14}$, respectively, are associated with $S = 0$.

**Table 4. Data collected by social scientist John Doe.**

| Case | C | R | L | E | S |
|------|---|---|---|---|---|
| $c_1$ | 1 | 0 | 1 | 1 | 0 |
| $c_2$ | 1 | 0 | 0 | 1 | 0 |
| $c_3$ | 1 | 0 | 0 | 1 | 0 |
| $c_4$ | 1 | 0 | 0 | 0 | 1 |
| $c_5$ | 1 | 0 | 0 | 0 | 1 |
| $c_6$ | 1 | 0 | 0 | 0 | 1 |
| $c_7$ | 1 | 1 | 1 | 0 | 1 |
| $c_8$ | 1 | 1 | 0 | 1 | 0 |
| $c_9$ | 1 | 1 | 0 | 1 | 0 |
| $c_{10}$ | 0 | 0 | 1 | 1 | 1 |
| $c_{11}$ | 0 | 0 | 0 | 1 | 1 |
| $c_{12}$ | 0 | 0 | 0 | 0 | 1 |
| $c_{13}$ | 0 | 1 | 1 | 0 | 0 |
| $c_{14}$ | 0 | 1 | 1 | 0 | 0 |
| $c_{15}$ | 0 | 1 | 0 | 0 | 0 |
| $c_{16}$ | 0 | 1 | 0 | 0 | 0 |

There are thus three alternative pathways to the outcome. For identifying the causal interplay between $R$, $C$, $L$, $E$ and $S$, Doe chooses the fs/QCA software [44], whose output is presented verbatim in Fig 2.

The solution is given as $C'R' + CE'$, a solution that is clear-cut yet unequivocally false. Instead of $CE'$, the presented result should have been the opposite, namely $C'E$; and instead of $C'R'$, the opposite in conjunction with $L$ should have been reported, namely $CRL$. Moreover, the third path of the DGS, $R'E'$, has not been discovered at all. Any interpretation of this solution that Doe were to propose would lead him further away from the truth rather than closer to it even though his data are ideal for discovering the DGS that social nature has predetermined. What went wrong?

```
File:  C:/Users/John Doe/QCA Research Project/my_data_for_qca.csv
Model:  S = f(C, R, L, E)
Algorithm:  Quine-McCluskey

--- TRUTH TABLE SOLUTION ---
frequency cutoff:  1
consistency cutoff:  1
Assumptions:
             raw        unique
           coverage   coverage    consistency
           ----------  ----------  ----------
~C*~R      0.428571    0.428571    1
C*~E       0.571429    0.571429    1
solution coverage:  1
solution consistency:  1
```

**Fig 2. Output of fs/QCA software for an analysis of Doe's data.**

That `fs/QCA` presented a solution that not even remotely reflects the DGS is no programming bug, nor an idiosyncratic scenario unlikely to be reproduced with different data, nor a fault attributable to QMC. Instead, it is a prime example of how far astray researchers can be led when methods developed in one discipline (electrical engineering) for one specific purpose (the minimization of hardware costs in architecting electrical switching circuits) are imported into another area (social sciences) without proper adaptation of these foreign methods to their new purpose (observational data analysis for causal inference). We have now everything in place to link Doe's research problem back to the previous section on QMC.

First notice that when Doe's data are transformed into a QCA truth table, this table is equivalent to the function table presented above in Table 1: simply replace the original factor label $V$ with the new label $C$, $W$ with $R$, $X$ with $L$, $Y$ with $E$ and $Z$ with $S$. Minterm 0 in Table 1 represents case $c_{12}$ in Table 4, minterm 1 case $c_{11}$, minterm 3 case $c_{10}$, minterm 8 cases $c_4$, $c_5$ and $c_6$, and minterm 14 case $c_7$. By extension, the prime implicant chart for Doe's data also presents exactly the same optimization problem as that presented in Table 3. The complete chart for Doe's data is provided in Table 5.

Recall the objective function specified in the previous section and QMC's approach to optimizing the process of identifying a switching function that would meet this objective function. The use of the principle of row dominance allowed QMC's to directly derive such a function without identifying any alternative irredundant sums that, under no circumstances, could outperform this function. Doe's interest, however, is in getting closer to the truth behind phenomenon $S$, not in building a low-cost electrical switching circuit. All that needs to be done to adapt QMC to the purpose of causal inference is to redefine the objective function such that it agrees with this purpose. As Graßhoff and May [22] had already demonstrated in the framework of propositional logic about two decades ago, any minimally necessary disjunction of minimally sufficient conjunctions is a potential candidate for a causal explanation under the INUS theory of causation. As propositional logic and switching circuit theory are completely equivalent branches of the same Boolean algebra, this simply means that, when translated to the terminology of electrical engineering, QMC must produce all irredundant sums, and not only a minimal sum, if employed for purposes of causal inference. Let us therefore reformulate $\mathcal{F}$ such that QMC should identify the set of all equivalent SOP functions that are irredundant, but not necessarily minimal.

Petrick's method provides a straightforward and well-established technique in electrical engineering that can be employed for finding all irredundant sums [41]. All we need is an auxiliary prime implicant function $p$. More specifically, let $C'E = P_1$, $CE' = P_2$, $C'R' = P_3$, $R'E' = P_4$ and $CRL = P_5$. After combining all prime implicants in a POS function such that each term is a sum of all prime implicants covering a positive minterm, the following transformations can be

**Table 5. Prime implicant chart for Doe's data presented in Table 4.**

| ID | | Minterms | | | | |
|---|---|---|---|---|---|---|
| | | 0 | 1 | 3 | 8 | 14 |
| | | $C'R'L'E'$ | $C'R'L'E$ | $C'R'LE$ | $CR'L'E'$ | $CRLE'$ |
| 1,3,5*,7* | $C'E$ | – | × | × | – | – |
| 8,10*,12*,14 | $CE'$ | – | – | – | × | × |
| 0,1,2*,3 | $C'R'$ | × | × | × | – | – |
| 0,2*,8,10* | $R'E'$ | × | – | – | × | – |
| 14,15* | $CRL$ | – | – | – | – | × |

carried out:

$$
\begin{aligned}
p &= (P_3 + P_4)(P_1 + P_3)(P_1 + P_3)(P_2 + P_4)(P_2 + P_5), \\
&= (P_3 + P_4)(P_1 + P_3)(P_2 + P_4)(P_2 + P_5) \qquad \text{by idempotency,} \\
&= (P_3 + P_4 P_1)(P_2 + P_4 P_5) \qquad\qquad\qquad \text{by distribution,} \\
&= P_2 P_3 + P_1 P_2 P_4 + P_1 P_4 P_5 + P_3 P_4 P_5 \qquad \text{by commutativity and idempotency.}
\end{aligned}
$$

Four irredundant sums exist for Doe's data. When $p$ is now transformed back, the following four models $m_1$ to $m_4$ emerge as candidates for explaining $S$:

$$
\begin{aligned}
m_1 : \quad S &= CE' + C'R', \\
m_2 : \quad S &= C'E + CE' + R'E', \\
m_3 : \quad S &= C'E + R'E' + CRL, \text{ and} \\
m_4 : \quad S &= C'R' + R'E' + CRL.
\end{aligned}
$$

Model $m_1$—the only model presented by `fs/QCA`—corresponds to the output of QMC given in the previous section, namely $V'W' + VY' = Z$. This, however, is not the DGS behind Doe's data. Instead, the correct model is $m_3$, which would have never been returned by QMC under sum minimality; nor would have been models $m_2$ and $m_4$, for that matter. Three out of four possible and equally well-fitting causal explanations of $S$ would therefore have never been brought to the attention of Doe. Instead, Doe would have been led to believe that the evidence is clear-cut and that the data allow of only a single explanation for $S$, which we know to be the wrong one.

In the following section, we analyze the extent to which this algorithmic bias has intruded into empirical work over the last quarter century. Concretely, we present a large-scale replication effort estimating the degree to which applied research in three social-scientific disciplines has been affected by algorithmic bias attributable to the use of sum minimality instead of sum irredundancy when employing Boolean optimization algorithms in QCA.

## Data

We drew on a total of 215 peer-reviewed articles from across 109 high-profile management, political science and sociology journals indexed in Clarivate Analytics' Journal Citation Reports. They are listed in S1 Appendix, and represent the final set out of a total of 357 eligible articles for which it was possible to gather sufficient information for re-running the analysis. Our sampling and data collection strategy, including the Preferred Reporting Items for Systematic Reviews and Meta-Analyses (PRISMA) flow diagram, can be found in S2 Appendix. The PRISMA checklist is included in S3 Appendix.

As many articles present several distinct solutions based on separate runs of QCA, the unit of analysis is not the study itself, but the presentation of a distinct solution. For instance, many studies present separate results for the presence and the absence of an outcome, or present analyses for different yet related outcomes. Across all 215 articles, 552 distinct QCA solutions turned out to be reanalyzable.

So as to minimize the risk of human errors, we automated the analysis via a purpose-built `R` function called `algoBias`, which imports the complete replication material at once, verifies the correctness and integrity of this material, and carries out the calculation of all bias statistics. This function also has an in-built emergency stop for data that generate so many equally well-fitting models that a meaningful interpretation becomes impossible. The function `algoBias` is available in the replication script, and can be easily re-used or adapted.

## Analytical strategy

In addition to algorithmic bias, empirical QCA studies may suffer from other types of bias. At least two additional, major types of bias that need to be factored out when computing the degree of bias solely attributable to algorithmic sources command attention. The source of the first lies in the discounting of evidence due to personal beliefs, a phenomenon referred to as confirmation bias in psychology [45, 46]. Although the negative consequences of preferring belief-compatible over belief-incompatible evidence have long been recognized, some QCA methodologists still explicitly recommend disregarding explanatory models that are not in accordance with theoretical preferences when faced with equally well-fitting counter-evidence. These recommendations seem to have given applied QCA researchers license to tacit model selection [30].

The second alternative source of bias consists in the use of so-called conservative (QCA-CS) and intermediate solutions (QCA-IS), which trade in the guarantee of methodological correctness of the parsimonious solution (QCA-PS) for a higher likelihood of generating stronger results via the back-door addition of artificial data [47–49]. We refer to the effect of this practice as data inflation bias.

The overall reporting bias in empirical QCA studies is thus an aggregate of at least three separate types of bias that need to be distinguished: confirmation bias, data inflation bias and algorithmic bias. The first two types simply need to be computed for identifying a study's degree of algorithmic bias, yet we will not pay closer attention to them here. More specifically, so as to isolate algorithmic bias, we reanalyzed each of the 552 QCA solutions under both sum minimality and sum irredundancy, and compared the obtained numbers of models to the number of models reported for a solution in the corresponding article. Concretely, let $m_{PS.SI}$ denote the total number of models fitting the data equally well under sum irredundancy in QCA-PS, $m_{PS.SM}$ the total number of such models under sum minimality in QCA-PS, $m_{CS.SI}$ the total number of such models under sum irredundancy in QCA-CS, $m_{CS.SM}$ the total number of such models under sum minimality in QCA-CS and $m_{rep}$ the number of models reported. Then, confirmation bias $b_{conf}$, data inflation bias $b_{dinf}$ and algorithmic bias $b_{algo}$ are computed as specified in Table 6.

As an illustration, imagine a study reported only a single model for a QCA-PS run on its set of data, but two models fit these data equally well under sum minimality and five models under sum irredundancy. Then, data inflation bias would be ruled out *a priori* because of the use of QCA-PS, confirmation bias would amount to $(2 − 1)/5 = 0.2$, algorithmic bias to $(5 − 2)/5 = 0.6$, and the overall reporting bias to $0.2 + 0.6 = 0.8$. Expressed as percentages, 20 per cent of all data-fitting models were consciously suppressed by the study's authors (confirmation bias), 60 per cent of all data-fitting models were unconsciously suppressed (algorithmic bias), and 80 per cent of all models fitting the study's data equally well were not reported.

**Table 6. Formulas for computing different types of bias in QCA.**

| Bias | QCA solution type | | | | |
|---|---|---|---|---|---|
| | QCA-CS | | QCA-IS | | QCA-PS |
| | $m_{PS.SM} \geq m_{CS.SM}$ | $m_{PS.SM} < m_{CS.SM}$ | $m_{PS.SM} \geq m_{CS.SM}$ | $m_{PS.SM} < m_{CS.SM}$ | |
| $b_{conf}$ | $\frac{m_{CS.SM} - m_{rep}}{m_{PS.SI}}$ | $\frac{m_{CS.SM} - m_{rep}}{m_{CS.SI}}$ | $\frac{m_{CS.SM} - m_{rep}}{m_{PS.SI}}$ | $\frac{m_{CS.SM} - m_{rep}}{m_{CS.SI}}$ | $\frac{m_{PS.SM} - m_{rep}}{m_{PS.SI}}$ |
| $b_{dinf}$ | $\frac{m_{PS.SM} - m_{CS.SM}}{m_{PS.SI}}$ | — | $\frac{m_{PS.SM} - m_{CS.SM}}{m_{PS.SI}}$ | — | — |
| $b_{algo}$ | $\frac{m_{PS.SI} - m_{PS.SM}}{m_{PS.SI}}$ | $\frac{m_{CS.SI} - m_{CS.SM}}{m_{CS.SI}}$ | $\frac{m_{PS.SI} - m_{PS.SM}}{m_{PS.SI}}$ | $\frac{m_{CS.SI} - m_{CS.SM}}{m_{CS.SI}}$ | $\frac{m_{PS.SI} - m_{PS.SM}}{m_{PS.SI}}$ |

## Results

The results of our meta-analysis are presented in Fig 3. The first fact to be noticed is that 361 out of all 552 analyzed solutions (about 65 per cent) are not affected by algorithmic bias. At the same time, that means every third published QCA solutions in our sample *is* affected, which is a considerable proportion. In other words, even when everyone involved in the production and release of scientific work—authors, reviewers, and journal editors—is assumed to have acted as ethically and objectively as they possibly could, and even when the analyzed data are assumed not to be beset by any other problems impacting negatively on the quality of the reported results, our analysis suggests that every third QCA solution presented in a management, political science or sociology journal has oversold its solution simply because the algorithm in the chosen QCA software operated under an objective function that is suitable for one specific purpose in electrical engineering applications, but not for causal inference.

When magnitude, rather than the mere presence or absence of bias, is taken into account, our results show some interesting nuances. For almost all QCA solutions that are affected by algorithmic bias (about 31 per cent overall), at least half of all viable candidate models have never been brought to the attention of researchers or the readers of these researchers' published article that presented this solution ($b_{algo} \geq 0.5$). If we increase the magnitude to $b_{algo} \geq 0.75$, this figure drops to about 17 percent. However, at the extreme end of $b_{algo} \geq 0.95$, which refers to all situations in which only a single model was reported by the authors although at least 20 models fit the data equally well, about one QCA solution in 12 (8.5 per cent) is still affected. Contained within this figure are no fewer than 36 solutions, amounting to 6.5 per cent of all presented QCA solutions, for which the number of models was so high that virtually nothing could have been inferred from the data.

An example study which did manage to avoid confirmation and data inflation bias but whose results suffer from an enormous amount of algorithmic bias is Stoiber and Töller [50]. This study seeks to analyze the causes of privatization of hospital order treatment of criminals with diminished capacity in the German Bundesländer by means of multi-value QCA and the software `Tosmana` (multi-value QCA has been employed far less than other QCA variants, but `Tosmana` has enjoyed a software market share of around 15 percent [51]). Five out of 16 Bundesländer have privatized such treatments. The authors acknowledge that solutions explaining privatization are ambiguous, but only a tiny fraction of this ambiguity is reported.

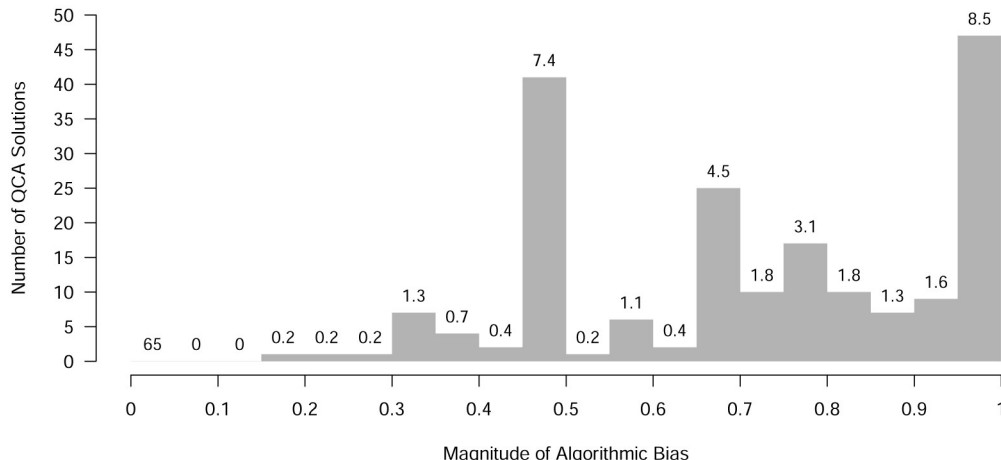

**Fig 3. Frequency of QCA solutions (overall percentage share above bars) along magnitudes of algorithmic bias (0: no bias, 1 complete bias; bar for "no bias" removed for layout reasons).**

More precisely, `Tosmana` presented three models to the authors, two of which contained the same two prime implicants. These two prime implicants together covered four of the five positive cases. As a result, the authors believed that their analysis led to a relatively clear-cut finding ("einem eindeutigen Ergebnis für drei Bundesländer [. . .]. Auch für Thüringen erbringt die sparsame Lösung einen eindeutigen Befund [. . .].")· To the readers of this study, it thus seems as if there exists slight model ambiguity, but the effects of this problem are attenuated by the existence of two shared prime implicants covering four of the five Bundesländer that implemented privatization. Yet, in actuality, there exist 72 other and equally well-fitting models, among which the supposedly central prime implicants are not central any more. What `Tosmana` reported were only the three minimal sums from among a total of 75 irredundant sums that should have been reported. Stoiber and Töller's study is thus commendable insofar as the authors objectively report all models presented to them by their chosen software, yet the enormous amount of algorithmic bias induced by `Tosmana`'s undocumented use of sum minimality instead of sum irredundancy unfortunately completely overshadows this positive aspect.

## Conclusion

In this article, we have revealed a type of bias that has gone almost unnoticed to date in the context of science's replication crisis: algorithmic bias. Concretely, we have shown why the uncritical import of the Quine-McCluskey algorithm (QMC) from electrical engineering to data analysis with the method of Qualitative Comparative Analysis (QCA) for purposes of causal inference has led to considerable algorithmic bias due to the simultaneous import of an objective function under which QMC operates in electrical engineering applications. This objective function, called *sum minimality*, is appropriate for architecting minimum cost switching circuits, but not for causal inference. For purposes of causal inference, *sum irredundancy* provides the correct objective function for any optimization algorithm, QMC or otherwise.

Why this misapplication of objective functions in QCA-based literature has gone unnoticed for decades remains an important question. We see two conditions that, together, may provide a significant part of the explanation: first, despite its heavy reliance on components originally developed in electrical engineering, QCA is a method that has evolved in sociology and political science, but hardly any social scientist takes courses in logic design or reads textbooks on electrical engineering. From this perspective, the borrowing of algorithms from disciplines which one has no aim to become familiar with is a highly risky business in the first place. Second, simulations were never implemented before or shortly after QCA had been released to the "market". While it is standard in many disciplines to test new algorithms in advance of their application to substantive problems, neither early critics nor proponents of QCA seem to have ever considered implementing appropriate tests to verify that sum minimality would be suitable for purposes of causal data analysis. Beyond the realm of QCA, our article should thus caution scientists more generally against letting elaborate methods and algorithms travel *too* easily across disparate disciplines without sufficient prior evaluation of their suitability for the context in hand. Sometimes, extensive upfront testing is preferable to post-release corrections.

## Supporting information

**S1 Appendix. List of included QCA articles.**
(PDF)

**S2 Appendix. Data collection strategy, including Preferred Reporting Items for Systematic Reviews and Meta-Analyses (PRISMA) flow diagram.**
(PDF)

**S3 Appendix. Preferred Reporting Items for Systematic Reviews and Meta-Analyses (PRISMA) 2009 Checklist.**
(DOC)

## Acknowledgments

Previous versions of this article have been presented at the conference "Scientific Integrity in Qualitative Research (SCIQUAL)", Utrecht University, the Netherlands, 13-14 September 2017; the 3[rd] annual conference of the Political Methodology section of the Political Studies Association, University of Essex, UK, 12 January 2018; the annual conference of the Swiss Political Science Association, University of Geneva, Switzerland, 5-6 February 2018; the "QCA Workshop" at the Cologne Center for Comparative Politics, University of Cologne, Germany, 25-26 April 2018; the annual conference of the Methods section of the German Political Science Association, Goethe University Frankfurt / Main, Germany, 4-5 May 2018; the VIII. European Congress of Methodology of the European Association of Methodology, Friedrich-Schiller-University Jena, Germany, 24-27 July 2018; the General Conference of the European Consortium for Political Research, University of Hamburg, Germany, 22-25 August 2018; the 3[rd] mid-term conference of the Quantitative Methods section of the European Sociological Association, Jagiellonian University Cracow, Poland, 3-6 October 2018; the Trilateral Conference of the Austrian, German and Swiss Political Science Associations, ETH Zurich, Switzerland, 14-16 February 2019; and the European Conference on Data Analysis, University of Bayreuth, Germany, 18-20 March 2019. We thank all conference participants who gave feedback, the two anonymous reviewers at PLOS ONE for their useful comments, and all authors who supported this project by sharing their data with us.

## Author Contributions

**Conceptualization:** Alrik Thiem, Tim Haesebrouck.

**Data curation:** Alrik Thiem, Lusine Mkrtchyan, Tim Haesebrouck, David Sanchez.

**Formal analysis:** Alrik Thiem.

**Funding acquisition:** Alrik Thiem.

**Investigation:** Alrik Thiem, Lusine Mkrtchyan.

**Methodology:** Alrik Thiem.

**Project administration:** Alrik Thiem.

**Software:** Alrik Thiem.

**Validation:** Alrik Thiem.

**Visualization:** Alrik Thiem.

**Writing – original draft:** Alrik Thiem.

**Writing – review & editing:** Alrik Thiem, Lusine Mkrtchyan, Tim Haesebrouck.

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
