## [Decision Letter · Decision Letter 0]

3 Feb 2020

PONE-D-19-32036

Algorithmic Bias in Social Research: A Meta-Analysis

PLOS ONE

Dear Prof. Dr. Thiem,

Thank you for submitting your manuscript to PLOS ONE. After careful consideration, we feel that it has merit but does not fully meet PLOS ONE’s publication criteria as it currently stands. Therefore, we invite you to submit a revised version of the manuscript that addresses the points raised during the review process.

As you can seen, both reviewers found your work interesting and technically sounding. However, I believe that the points raised by reviewers are important and should be thoughtfully addressed/discussed in a new version of the manuscript. 

We would appreciate receiving your revised manuscript by Mar 16 2020 11:59PM. To enhance the reproducibility of your results, we recommend that if applicable you deposit your laboratory protocols in protocols.io, where a protocol can be assigned its own identifier (DOI) such that it can be cited independently in the future. For instructions see: http://journals.plos.org/plosone/s/submission-guidelines#loc-laboratory-protocols

We look forward to receiving your revised manuscript.

Kind regards,

Sandro Meloni, Ph.D.

Academic Editor

PLOS ONE

Journal Requirements:

Reviewers' comments:

Reviewer's Responses to Questions

**Comments to the Author**

1. Is the manuscript technically sound, and do the data support the conclusions?

Reviewer #1: Yes

Reviewer #2: Partly

2. Has the statistical analysis been performed appropriately and rigorously? 

Reviewer #1: Yes

Reviewer #2: Yes

3. Have the authors made all data underlying the findings in their manuscript fully available?

Reviewer #1: Yes

Reviewer #2: Yes

4. Is the manuscript presented in an intelligible fashion and written in standard English?

Reviewer #1: Yes

Reviewer #2: Yes

5. Review Comments to the Author

Reviewer #1: I read with interest the paper Algorithmic Bias in Social Sciences. In this paper, the authors present a problem in the QMC algorithm that is traditionally used to perform QCA. This algorithm only presents part of the results, in the sense that there are results that equally fit the data but are not reported because of the way the algorithm is constructed. In re-analyzing data from a little more than 200 QCA studies from various social sciences (political science, management and sociology), they find that at least than 1/3 probably rely on the problematic algorithm because they do not repot all results that fit the data.

Full disclosure: I reviewed the paper for another journal last year. In the meantime, it seems that the authors have integrated some of my comments. Yet they have not addressed all of them. In this review, I will thus repeat the comments that have not been integrated, as well as new comments that emerged after another reading the paper.

I think the paper is of great quality: the argument and demonstration make a lot of sense, it is competently and convincingly written, and the data collection effort for the re-analysis is impressive. Yet I think the author should consider the following comments before its publication in PLOS ONE.

1. The authors can do a better job at drawing all implications from their study.

In re-analyzing datasets from published articles, the authors find that a QCA performed with the right algorithm frequently leads to multiple results (for a single dataset). These results are equivalent in the sense that they equally fit the dataset. They argue that it is then impossible for researchers to know which one is a better reflection of the data-generating structure, and that they should thus report all results in noting that they can all be the data-generating structure. In other words, after most QCA, researchers should say “maybe this result is the right one, or maybe it is this other one, I don’t know”. From Figure 3, we see that it isn’t rare that the right QCA algorithm produces more than 10 different results. This finding has important practical implications for QCA users. Assuming that they are looking for the data-generating process, it seems that the risk of not being able to identify it among other potential data-generating processes is very real. This seems to me as yet another case against QCA.

In view of this, I have two suggestions:

First, I am encouraging the authors to acknowledge that the possibility of having multiple results fitting equally well the dataset can reduce the appeal of QCA for social scientists.

Second, I am encouraging the authors to discuss the conditions under which QCA with the right algorithm can unequivocally find the data-generating structure (number of cases, number of conditions…). This would be of practical use for QCA users and would somehow balance the case against QCA discussed above.

2. I was wondering what the evolution of the algorithm problem pointed by the authors. It would be interesting to report a figure that shows the proportion of problematic QCA papers by decades. I suspect that the problem is less frequent now than 30 years ago because QCA users increasingly use the QCA R package that uses the right algorithm presented in this paper.

3. It would be interesting to reflect (perhaps in the conclusion) about the publication bias in QCA. As we all know, not all scientific research gets published. Hence, what the authors observe in their re-analysis of published articles is only a subset of what is happening in QCA research. And the set of published articles is probably different from the set of unpublished studies in many ways. Do they authors have any idea about of how different their result would have been if they had accessed to these unpublished studies? In other words, do they think that the algorithm problem in QCA is more severe or less severe in unpublished studies? My impression is that the problem is less severe in unpublished studies because the studies that report multiple possible results are harder to published than those with only one result (journals and reviewers liking neat results). In other words, there might be more QCA using the right algorithm among the unpublished studies. I’m curious to hear what the authors think about that.

Reviewer #2: This is an important paper in which the authors address a neglected methodological issue arising in using Qualitative Comparative Analysis (QCA). As it stands, the paper has various shortcomings which ought to be addressed before the paper is ready for publication. They broadly fall into two areas, technical points and implications for wider issues in social science research.

Technical points

It would be helpful if the authors could comment on the role of limited diversity in algorithmic bias. Is it likely to exacerbate the problem, or is algorithmic bias unrelated to limited diversity? Are there any other features of the data which may make algorithmic bias more or less likely?

The sections “Switching Circuit Optimization with QMC: A Primer” and “Algorithmic Bias” are clearly written and very valuable indeed. The authors’ own empirical analysis is carried out competently given the sample used, but they should make it clear in the main body of the text that the 214 papers used constitute only around 60% of the 360 papers deemed “eligible”. This information is currently only to be found in an online appendix. Accordingly, the authors should moderate their claims regarding the likely extent of the problem of algorithmic bias, given that they cannot be sure whether their 214 papers are representative of all published papers in this area.

The authors should explain how they arrived at their method of computing publication bias (pp.15/23). If they did so, it might become clearer why a compound measure of three separate types of bias is needed or helpful – this is currently not clear. All three types of bias included in the measure are complex and it may not be possible (or sensible) to capture them within a single indicator. Confirmation bias is a real phenomenon, but the authors do not allow for the possibility that a researcher’s preference for a particular model may actually be justified because it is based on sound theoretical analysis. Inflation bias can only arise if there is limited diversity in the data, so it is not relevant to all studies. Both confirmation bias and inflation bias are important topics in their own right, but without detailed discussion and analysis, their inclusion in the overall measure of publication bias seems slightly arbitrary, not least because there are likely to be other sources of publication bias in addition to the three included here. If the authors do wish to measure algorithmic bias, a more transparent way of doing so would be to develop a single, one-dimensional measure for it.

Implications for wider issues in social science research

Linked to the last point I made under “technical issues” is a wider issue concerning conceptual clarity, which is that algorithmic bias is actually not the same thing as publication bias. Publication bias arises from a tendency of submitting to and acceptance by journals of positive results only. The authors themselves acknowledge that these are different points in their introduction where they note that publication bias arises from selective reporting which is committed consciously, unlike algorithmic bias of which authors are not even aware. So while publication bias is an important topic, the term should not be widened unnecessarily.

The authors claim that algorithmic bias contributes to the “reproducibility crisis” without any supporting evidence. They certainly do not provide any evidence as to why algorithmic bias “may be one of the largest contributors to irreproducible research and publication bias in some areas of science”, p.2/23, thus this appears to be mere speculation.

Likewise, the authors claim that problems arise when methods travel across disciplinary boundaries. Indeed, this appears to be the case with algorithmic bias which only arises in a social science context but which is irrelevant in the context of electric engineering where the Quine-McCluskey algorithm was first developed. But the authors provide no analysis of why such problems should arise more generally whenever methods travel across disciplinary boundaries. Granted, a careful analysis of this possibility should be undertaken when methods are employed in a novel context, but the danger with the authors’ general warning is that researchers may become adverse to employing methods from other areas altogether. But this would potentially deprive them of valuable innovative tools.

Similarly, the paper’s title itself is misleading: algorithmic bias is a serious problem well worth the authors’ thorough discussion, but it is not a problem for all of social science.

6. PLOS authors have the option to publish the peer review history of their article (what does this mean?). If published, this will include your full peer review and any attached files.

Reviewer #1: No

Reviewer #2: No

---

## [Decision Letter · Decision Letter 1]

10 Apr 2020

PONE-D-19-32036R1

Algorithmic Bias in Social Research: A Meta-Analysis

PLOS ONE

Dear Prof. Dr. Thiem,

Thank you for submitting your manuscript to PLOS ONE. After careful consideration, we feel that it has merit but does not fully meet PLOS ONE’s publication criteria as it currently stands. Therefore, we invite you to submit a revised version of the manuscript that addresses the points raised during the review process.

As you can see, both reviewers believe that with this resubmission the work has substantially improved and only have minor remarks. My decision is "Minor Revision" to give you the opportunity to address them and, at least, discuss point 2 of Reviewer 2.  

We would appreciate receiving your revised manuscript by May 25 2020 11:59PM. To enhance the reproducibility of your results, we recommend that if applicable you deposit your laboratory protocols in protocols.io, where a protocol can be assigned its own identifier (DOI) such that it can be cited independently in the future. For instructions see: http://journals.plos.org/plosone/s/submission-guidelines#loc-laboratory-protocols

We look forward to receiving your revised manuscript.

Kind regards,

Sandro Meloni, Ph.D.

Academic Editor

PLOS ONE

Reviewers' comments:

Reviewer's Responses to Questions

**Comments to the Author**

1. If the authors have adequately addressed your comments raised in a previous round of review and you feel that this manuscript is now acceptable for publication, you may indicate that here to bypass the “Comments to the Author” section, enter your conflict of interest statement in the “Confidential to Editor” section, and submit your "Accept" recommendation.

Reviewer #1: All comments have been addressed

Reviewer #2: (No Response)

2. Is the manuscript technically sound, and do the data support the conclusions?

Reviewer #1: Yes

Reviewer #2: Partly

3. Has the statistical analysis been performed appropriately and rigorously? 

Reviewer #1: Yes

Reviewer #2: Yes

4. Have the authors made all data underlying the findings in their manuscript fully available?

Reviewer #1: Yes

Reviewer #2: Yes

5. Is the manuscript presented in an intelligible fashion and written in standard English?

Reviewer #1: Yes

Reviewer #2: Yes

6. Review Comments to the Author

Reviewer #1: I'm satisfied with the way the author addressed my comments.

I've only one small request before the paper is published:

In their response to my first comment, the authors say "The fact that multiple explanations often fit a set of data is not unique to QCA, but also exists in regression analysis and experiments (e.g., Eberhardt 2013)". That's intriguing. I didn't know about that, and I think most researchers using regressions and experiments don't know either. It would be very helpful if the author could add a footnote in the main text to explain the intuition behind this claim.

Reviewer #2: The revision has addressed most of the points raised by myself and the other reviewer, and the paper is much improved. There are just three points which the authors have not addressed satisfactorily:

1. The information on the number of eligible papers does need to be in the main body of the text (my previous comment 2). Contrary to what the authors say, this is not related to reviewer 1’s comment 3 but it is a separate point: The authors have identified a population of 360, and they have been able to analyse 215 out of these 360 papers. The reasons they give for not analysing them all are perfectly acceptable, but they claim that “our analysis suggests that every third QCA solution presented in a management, political science or sociology journal has oversold its solution simply because the algorithm in the chosen QCA software operated under an objective function that is suitable for a specific purpose in electrical engineering applications, but not for causal inference” (p.17). Readers should be aware that this claim concerning “every third QCA solution” rests on a sample roughly 60% the size of the population identified by the authors.

2. Adding a reference has not really addressed the point concerning reproducibility (my previous comment 5). There would have needed to be evidence that authors try and fail to replicate findings based on QCA in the first place (I am not aware that there are many attempts to do this), before analysing whether algorithmic bias is to blame. Given that the algorithmic bias discussed in this paper arises only in the context of QCA, it is hard to see how it could contribute to irreproducible findings elsewhere.

3. The title is misleading (my previous comment 6), contrary to the authors’ claim. Algorithmic bias is an important problem, but it is one specific to QCA which in turn is one of many methods employed within social science. The fact that other problems “may” exist in the social sciences does not change this.

7. PLOS authors have the option to publish the peer review history of their article (what does this mean?). If published, this will include your full peer review and any attached files.

Reviewer #1: No

Reviewer #2: No

---

## [Editor Report · Decision Letter 2]

11 May 2020

Algorithmic Bias in Social Research: A Meta-Analysis

PONE-D-19-32036R2

Dear Dr. Thiem,

We are pleased to inform you that your manuscript has been judged scientifically suitable for publication and will be formally accepted for publication once it complies with all outstanding technical requirements.

With kind regards,

Sandro Meloni, Ph.D.

Academic Editor

PLOS ONE
---

## [Editor Report · Acceptance letter]

22 May 2020

PONE-D-19-32036R2

Algorithmic bias in social research: A meta - analysis

Dear Dr. Thiem:

I am pleased to inform you that your manuscript has been deemed suitable for publication in PLOS ONE. Congratulations! Your manuscript is now with our production department.

With kind regards,

on behalf of

Dr. Sandro Meloni

Academic Editor

PLOS ONE